# The effect of modifier and a water-soluble fertilizer on two forages grown in saline-alkaline soil

Shengchen Zhao[1,2], Dapeng Wang[1,2‡], Yunhui Li[3], Wei Wang[3], Jihong Wang[1,2], Haibo Chang [1,2]*, Jingmin Yang[1,2]*

**1** College of Resource and Environmental Science, Jilin Agricultural University, Changchun, Jilin Province, China, **2** Key Laboratory of Saline-Alkali Soil Improvement and Utilization, Ministry of Agriculture and Rural Affairs, Beijing, China, **3** College of Engineering, Jilin Normal University, Siping, Jilin Province, China

‡ DW contributed equally to this work and is co-first author of the paper.
* changhb@jlau.edu.cn (HC); yangjingmin@jlau.edu.cn (JY)

**Data Availability Statement:** All relevant data are within the manuscript and its Supporting information files.

## Abstract

Saline-alkali soil significantly impairs crop growth. This research employs the impacts of the modifier and water-soluble fertilizer, as well as their interaction, on the root systems of alfalfa and leymus chinensis in saline-alkali soil. The results exhibit that the hydrochar source modifier effectively enhances the root growth of both forage species. There are certain improvements in the root growth indicators of both crops at a dosage of 20 g/kg. Root enzyme activity and rhizosphere soil enzyme activity are enhanced in alfalfa, showing significant improvements in the first planting compared to the second planting. The application of water-soluble fertilizers also promotes root growth and root dehydrogenase activity. The root dehydrogenase activity of alfalfa and leymus chinensis are enhanced 62.18% and 10.15% in first planting than that of blank, respectively. Additionally, the two-factor variance analysis revealed a correlation between rhizosphere soil enzyme activity and changes in root traits. Higher rhizosphere soil enzyme activity is observed in conjunction with better root growth. The combined application of a modifier and water-soluble fertilizer has demonstrated a significant interaction effect on various aspects of the first planting of alfalfa and leymus chinensis. Moreover, the combined application of the modifier and water-soluble fertilizer has yielded superior results when compared to the individual application of either the modifier or the water-soluble fertilizer alone. This combined approach has proven effective in improving saline-alkali soil conditions and promoting crop growth in such challenging environments.

## Introduction

Land salinization is a global issue that adversely affects agriculture, the environment, and economic development [1]. The total area of saline-alkali soil with broad range, large area, and shaky structure in the world is approximately 95,438 million hm$^2$, which mainly concentrated in Eurasia, Africa, western America, and other regions. The distribution of saline-alkali soil

**Funding:** This study was supported by the Science and Technology Department of Jilin Province (20200201011JC) and Education Department of Jilin Province (JJKH20220343KJ).

**Competing interests:** The authors have declared that no competing interests exist.

varies across different countries and regions due to variations in geographical location and climatic conditions [2]. The Songnen Plain with an area of 37,000 km$^2$ is the largest saline-alkali land in China. Among the three major regions worldwide where salt and alkali are distributed, the Songnen Plain demonstrates the highest ecological degradation rate. Land salinization in arid and semi-arid climate zones presents a significant threat to sustainable agricultural development [3, 4]. According to literature data, saline-alkali soil is expanding with a rate of 0.1– 0.15 million hm$^2$ [5]. Land salinization leads to the degradation of soil physical properties, resulting in increased soil bulk density and reduced air permeability. Additionally, it causes nutrient deficiency and accumulation of salt in saline-alkali soil, leading to water pollution [6, 7]. Salinity stress causes disturbance in plant physiological status [8, 9] and imbalance nutrient absorption [10, 11], causing week growth and low crop productivity and quality [12, 13]. In order to ensure the availability of arable land and achieve consistent growth and stability in grain production, it is imperative to enhance and harness the potential of high-potential saline-alkali soil [14]. Indeed, selecting appropriate land use patterns that align with regional environments, ecological characteristics, and production requirements is crucial. In Central Europe, for instance, the development of pastures has been found to effectively maintain soil quality by providing natural protection against the significant land use system changes that result from agricultural intensification and urbanization. By avoiding such large-scale changes, the reduction of soil quality can be minimized [15, 16]. Therefore, pasture development offers several advantages over traditional agricultural production in saline-alkali land areas with harsh environmental conditions. Among these advantages, alfalfa stands out due to its strong stomatal tolerance and better salt tolerance compared to other species. It can thrive in moderate salt and alkali conditions and can be cultivated using a compact planting pattern. Additionally, alfalfa seedlings establish quickly, covering the land and reducing soil moisture evaporation effectively. This vegetation cover also aids in preventing salt accumulation on the soil surface [17–19]. Leymus chinensis also has a strong ability to tolerate saline-alkali stress as well as resist grazing pressure [20]. The enhancement of saline-alkali grasslands holds great significance for both the ecological environment and agricultural development on a global scale.

Saline-alkali land can be modified through a combination of physical, chemical, and biological methods. While physical improvement methods are effective, they often come with high costs. On the other hand, biological improvement methods offer a more cost-effective alternative with a longer improvement cycle. And chemical improvement methods are simple and effective which are popular methods for improving saline-alkali soil. Soil modifiers are preparations specifically designed to enhance soil structure and improve physical and chemical indexes. Upon application, soil modifiers facilitate the formation of a stable soil aggregate structure and soil amine complex. Consequently, these improvements contribute to enhanced soil fertility and increased crop yield [21]. Water-soluble fertilizers possess the advantageous characteristics of swift and effortless dissolution in water, allowing for a high utilization rate through effective absorption by vegetation. The medium element water-soluble fertilizer contains a high concentration of $Ca^{2+}$ and $Mg^{2+}$ which replaced with $Na^+$ in saline-alkali soil to reduce its harm [22]. The addition of modifier can reduce the soil pH value. And a large amount of $Ca^{2+}$ and $Mg^{2+}$ promote crop root growth on saline-alkali soil [23]. The root channel is an essential component of soil macropores and exudates of root can increase soil enzyme activity [24, 25]. Indeed, the growth of the root system can play a significant role in improving saline-alkali soil. As vegetation roots penetrate and develop within the soil, they can contribute to the amelioration of the soil environment.

It has been documented that amending soils, especially that suffering from environmental stresses, has a favorable effect on different soil properties [26–28]. Thus, soil productivity and

crop quality were improved by exploiting the recycled plant residues [29–31]. Biochar is a novel soil modifier method for saline-alkali soil that significantly improves soil nutrients and properties [32, 33]. Several studies have shown that biochar has a greater improvement effect on saline-alkali soil compared to straw, attributed to its surface's rich functional groups that can absorb and enhance nutrient holding capacity [6, 34]. Because of its porosity, biochar can improve soil physical structure [35], promote seed germination and crop growth [36], and improve soil enzyme activity and remove soil pollutants. Biochar itself is rich in carbon, and its ashes contains mineral elements needed for plant growth, which can improve soil permeability and increase soil carbon stock. In addition, the oxygen-containing functional groups on the surface of biochar, such as -COOH, -OH, etc., can react with the salt ions adsorbed in the soil colloid and adsorb $Na^+$, which is the most representative of saline and alkaline soils, and thus improve the salinity and alkalinity of the soil, therefore, biochar has a broad application prospect in soil improvement [37, 38]. Application of water-soluble fertilizers improves soil quality and increases crop yields in alkaline soils [39].

Hydrothermal carbon is gaining popularity due to its lower preparation temperature and energy consumption compared to the high temperature preparation method. During this process, a significant amount of organic acid is produced, which can modify saline-alkali soil. Currently, the combination of hydrochar, organic acid (liquefaction product of hydrochar), and water-soluble fertilizer to improve saline-alkali soil is still uncommonly studied.

In this study, soil modifiers were prepared using hydrochar, mushroom bran, humic acid, and fine sand. The water-soluble fertilizer containing medium elements was prepared using the liquefaction product of hydrochar and inorganic salt. The study will evaluate the improvement effect on saline-alkali soil by applying the soil modifiers and water-soluble fertilizer. Our study aims to investigate the effects of soil modifiers and medium element water-soluble fertilizer on crop root growth and rhizosphere soil enzyme activities, as well as the interaction between the soil modifier and water-soluble fertilizer. This research offers a promising strategy for the sustainable utilization of straw and the effective improvement of saline-alkali soil.

## Materials and methods

### Materials

The saline-alkali soil sample was collected in Erlong Town, Taonan City, Jilin Province, which belonged to Songnen Plain. It was taken in mid-July of 2021. The soil depth ranged from 0 to 20 cm. It was sifted through 2 mm sieve after thoroughly mixed and dried. The basic physical and chemical indexes of the saline-alkali soil sample were displayed in S1 Table in S1 File.

The rice straw was obtained from the rice production field of Agricultural Science Experimental Station of Jilin Agricultural University. It was naturally dried, crushed and sifted through a 60-mesh sieve. The Auricularia auricula fungus waste chaff was collected in Dapu Chaihe Town, Dunhua City, Jilin Province. The component of the original the medium was wood sawdust (85%), wheat bran (10%), soybean cake powder (3%), gypsum (1%), and quicklime (1%). The waste fungus chaff was crushed and mixed to prepare the modifier, which pH value was 7.13. The nutrient content of the rice straw and the waste fungus chaff were shown in S2 Table in S1 File.

Humic acid purchased from Harbin Yishida Ecological Technology Development Co., Ltd. The content of effective humic acid was 71.85%, and pH was 4.42.

The seeds of alfalfa got from Ningxia, and the seeds of leymus chinensis was gained from Jilin. Before field micro area experiment, two kinds of grass seeds were selected and then incubated in constant temperature and light incubator. The germination rate of alfalfa seeds and leymus chinensis seeds were 78.5% and 97%, respectively.

## Preparation method of soil modifier and water-soluble fertilizer

The rice straw was crushed and mixed with deionized water. After 30 minutes of stirring at room temperature, it was transferred into a 1000 ml Teflon-lined stainless steel autoclave and heated at 240˚C for 24 hours under autogenous pressure [1, 35]. The hydrochar was filtered and divided into solid and liquid parts to prepare the soil modifier and water-soluble fertilizer, respectively. The C/N atomic ratio of hydrochar was 29.36. The liquid component was acid mixed solution with pH = 4.46.

The soil modifier was prepared by hydrochar, mushroom bran, humic acid, fine sand, and liquid component with mass ratios 0.5: 4.5: 2: 1: 2. Stirred the mixture until homogeneous mixing and adjusted pH value of solution to 6.50 with dilute hydrochloric acid. The soil modifier contained 20% water and 42.92 g/kg organic matter that was denoted M in experiments (The amount of soil modifier from 10 g/kg—50 g/kg were marked by M10, M20, M30, M40, and M50, respectively). The blank experiment was carried out under the same conditions without soil modifier that was marked M0.

The water-soluble fertilizer was obtained by liquid component, calcium nitrate tetrahydrate and magnesium nitrate hexahydrate. According to the technical standard of medium element water-soluble fertilizer (NY 2266–2012), the content of medium element should be equal or greater than 100 g/L. As a result, by adding calcium nitrate tetrahydrate and magnesium nitrate hexahydrate in the medium element water-soluble fertilizer in this experiment, the content of calcium and magnesium reached 60 g/L and 40 g/L, respectively. The water-soluble fertilizer would be diluted and sprayed on leaves. The volume ratio of water-soluble fertilizer and water was 1:1500. The pH value of the water-soluble fertilizer was 6.95. On basis of the amounts of modifiers were 10 g/kg—50 g/kg (In conjunction with the application of water-soluble fertilizer), which were marked with MF10, MF20, MF30, MF40, and MF50. The blank experiment was performed under the same conditions with only water-soluble fertilizer without modifier that was marked MF0.

## Method of field micro area experiment

This experiment was conducted from 10th May to 20th September 2021 in Erlong Village, Erlong Township, using in situ simulation of micro-area trial field. The land was divided into 138 test plots in the form of 1.00m × 1.00m. The soil of each test site was pre-treated, the top layer of tillage soil (above 20.00 cm) was removed from each area and mixed with modifiers (amount of modifier from 10 g/kg to 50 g/kg) and then refilled back to its original position to ensure that the physicochemical properties were basically the same, and 100 seeds were sown uniformly in each sub-district. The crop was sprayed with water soluble fertilizer for the first time when it entered the three-leaf stage, and then sprayed every 10 days for a total of three times at 100 ml. After 50 days of growth, alfalfa and leymus chinensis were taken out of the soil with their roots, cleaned, and then brought back to the laboratory for the determination of the indicators. Alfalfa growth to 50 days, can be more obvious to see the difference between the different treatments of alfalfa growth, and in the alfalfa growth of 50 days when the unit alfalfa is small, and will not be affected by the amount of soil and growth, so the time of this test is set at 50 days. A second trial was conducted in the original soil on 1st August, the second trial no longer added modifiers only water-soluble fertilizer, other treatments were the same.

## Measuring indicators and methods

After a period of 6 days from planting, the number of seedlings for both alfalfa and leymus chinensis was recorded. After 50 days, all the forage plants were removed from the soil, and the number of surviving plants was counted. The soil adhering to the root surface was carefully

removed, and the forage plants were placed on a sieve, cleaned with water, and excess water was absorbed using filter paper. The forages from each treatment were stored separately and kept intact for further analysis.

Root dehydrogenase activity (DHA) was determined by dehydrogenase assay kit. The method was carried out according to the kit instructions, based on TTC colorimetric method [40]. The Content of malondialdehyde content (MDA) in roots was determined by malondialdehyde content determination kit, based on thiobarbituric acid colorimetry [41]. The total root length, total root surface area, average root diameter, and total root volume for the two kinds of forages were measured by the root scanning system.

The soil catalase activity (CAT) was determined by the kit with biochemical method. The specific steps were carried out in accordance with the instructions of the kit. An enzyme activity unit was defined by the degradation of 1μmol $H_2O_2$ per gram of soil sample per hour [42, 43]. The activity of soil alkaline phosphatase (ALP) was measured using a kit with biochemical method. The specific measurement procedures were performed in accordance with the instructions provided with the kit. Based on the disodium phenyl phosphate colorimetric method [44], 1 nmol p-nitrophenol (PNP) was produced by hydrolysis of PNP per gram of soil sample per hour as an enzyme activity unit. Soil sucrase activity (SC) was determined by kit with biochemical method. The specific experiment steps were obtained according to the kit instructions. Based on the 3,5-dinitrosalicylic acid colorimetric method [44], 1 mg glucose was produced by per gram of soil sample in 24 h as an enzyme activity unit. Urease activity (URE) was determined by phenol sodium-sodium hypochlorite colorimetric method [44], expressed as the number of milligrams of $NH_3$-N per gram of soil sample at 24 h.

### Statistical analysis

The data collation and chart drawing were carried out by Excel 2013 and Origin 2018. SPSS Statistics 20.0 was adopted for two factors analysis of variance ANOVA. R software vegan and ggplot 2 packages were used for redundancy analysis.

## Results

### Effects of modifier and water-soluble fertilizer on alfalfa

**Survival number.** S1A and S1B Fig in S1 File depict survival rate of the alfalfa under various modifier dosages. The survival rates of alfalfa in the two batches show a trend of first rising and then falling. The survival number of alfalfa with M20 treatment significantly increased which is higher 28.99% than that of M0 treatment in first planting (S1A Fig in S1 File). MF20 is the highest survival number of alfalfa 37 plants/ pot. Errors bars represent standard errors. Different lowercase letters above the bars indicate significant differences among treatments at P<0.05 with applied different amounts of modifier, which applied all the figures.

**Root biomass.** It can be seen from S2A and S2B Fig in S1 File, the root biomasses of alfalfa per plant in two batched are up with increasing amounts of modifier and water-soluble fertilizer (M0—M20 and MF0—MF20). Compared to the M0 treatment, the root biomass of alfalfa with M20 treatment obviously increases by 208.17% (P<0.05) and 43.75% (P<0.05), respectively. The addition of water-soluble fertilizer further improves the root biomass of alfalfa by 37.19% (P<0.05) in the first planting of alfalfa with M20 treatment. Nevertheless, no obvious effect is observed in the second planting of alfalfa.

**Root growth.** S3 Fig in S1 File displays the effects of modifier and water-soluble fertilizer on root growth of alfalfa. The root growth of alfalfa in two batches of alfalfa treated with M20 treatment are 43.23 cm and 74.00 cm, which are 17.35% (P<0.05) and 9.33% (P<0.05) higher

than that of treated with M0 treatment, respectively (S3(1) Fig in S1 File). The root length of alfalfa with MF20 treatment in first planting is 13.52% (P<0.05) than that of in MF0 treatment which due to adding water-soluble fertilizer (S3(2) Fig in S1 File). The total root surface area of alfalfa with MF20 treatment in two batches are higher than that of MF0 treatment by 16.51% (P<0.05) and 23.31% (P<0.05), respectively (S3(3) Fig in S1 File). The average root diameter and root volume of alfalfa with M20 treatment are significantly increase than that of M0 treatment 28.49% (P<0.05) and 26.32% (P<0.05) in first planting, respectively (S3(4) Fig in S1 File). The growth indexes of the root system obviously increase after added modifier. And the indexes generally reach their maximum when the modifier dosage is less than 20 g/kg.

**Root enzyme activity and rhizosphere soil enzyme activity.** The influence of modifier and water-soluble fertilizer on root dehydrogenase activity of alfalfa are shown in Fig 1A and 1B. The root dehydrogenase activity of alfalfa with M20 treatment in two batches have noticeable increases than that of with M0 treatment (62.18% (P<0.05) and 48.72% (P<0.05), respectively.). Different from the root growth index, with the increase of the amounts of modifier, the root dehydrogenase activity of alfalfa in the second planting displays a trend of rise to decline and then to rise, and with M50 treatment is 47.73% higher than that of M0 treatment (P<0.05), and the application of water-soluble fertilizer further enhance the activity by 5.80% (P<0.05). The trend has displayed in Fig 1B.

After the application of modifier, the change trends of malondialdehyde content in the roots of alfalfa in the first planting are tendency to rise, fall and rise again, and the content of malondialdehyde in M20 and M50 treatments is 9.15% (P<0.05) and 23.89% (P<0.05) lower than that of M0 treatment, respectively. Malondialdehyde content in roots of alfalfa in the second planting decrease first and then raise with the increase of the amount of modifier. M20 treatment significantly decrease the malondialdehyde content in roots, which is 12.57% lower than that of M0 treatment (P<0.05) (Fig 2A and 2B).

The modifier and water-soluble fertilizer also impact the enzyme activity of alfalfa rhizosphere soil. The experiment result illustrates that with M20 treatment increase the alkaline phosphatase activity, sucrase activity, and urease activity of rhizosphere soil by 47.02%, 27.45%, and 18.41%, respectively (Fig 3A and 3B). Compared to M0 treatment, there is no significant improvement on the catalase activity of alfalfa rhizosphere soil after applied modifier.

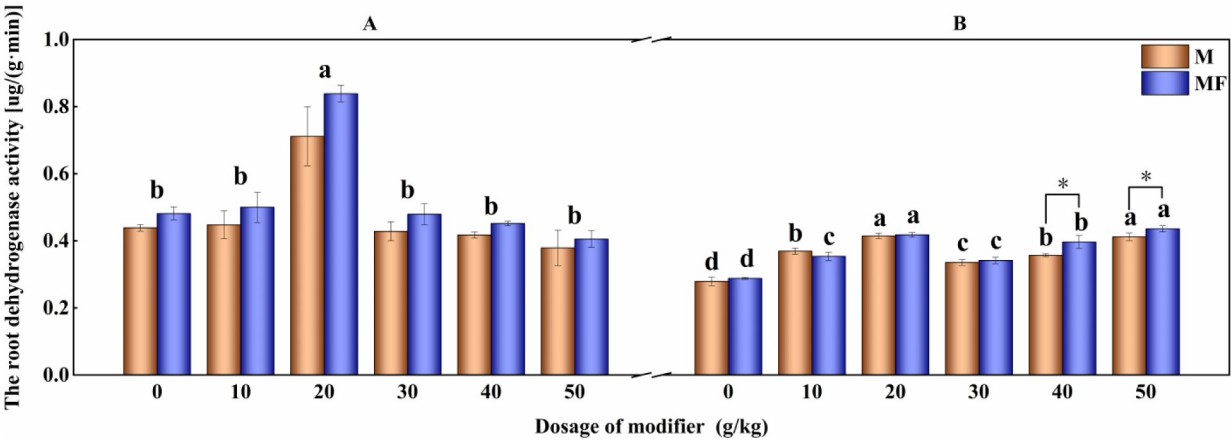

**Fig 1. Effects of modifier and water-soluble fertilizer on dehydrogenase activity of alfalfa roots.**

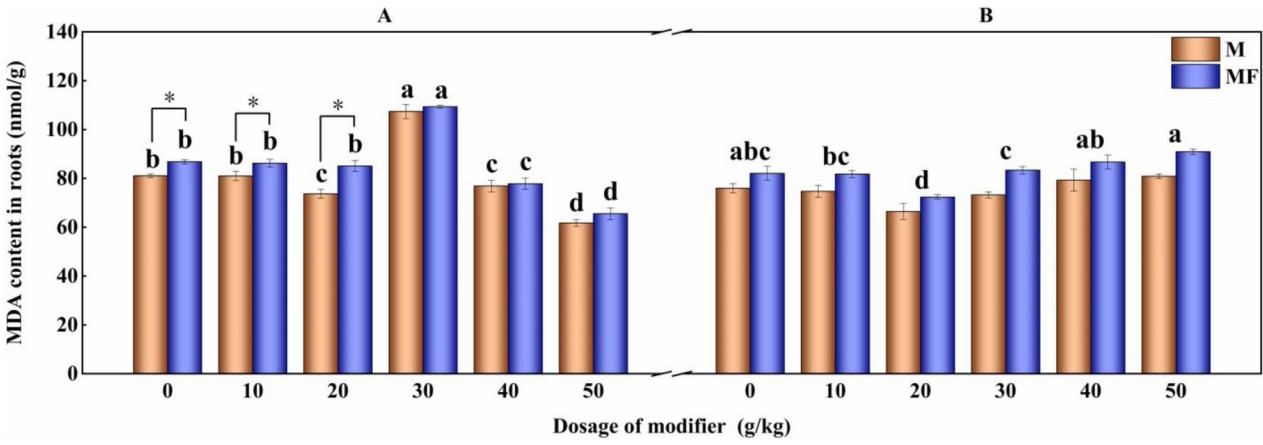

**Fig 2. Effect of modifier and water-soluble fertilizer on content of MDA in alfalfa roots.**

## Effects of modifier and water-soluble fertilizer on leymus chinensis

**Survival number.**   S4A and S4B Fig in S1 File depict the influence of modifier and water-soluble fertilizer on the survival number of leymus chinensis. With the increase of modifier dosage, the survival number of two batches of leymus chinensis raise first and then decrease. The survival number of leymus chinensis with M20 treatment significantly enhance 3.47% (P<0.05) and 17.59% (P<0.05) than that of M0 treatment.

**Root biomass.**   Compared to those of M0 treatment, the root biomass of leymus chinensis with M20 treatment in the first planting reach the highest (1.87 mg) which is 96.88% (P<0.05). It is elevated 13.04% (P<0.05) after adding water-soluble fertilizer (S5A Fig in S1 File).

**Root growth.**   The total root of length with M20 treatment in first planting reach obviously 269.54 cm which higher 12.22% (P<0.05) than that of M0 treatment (S6(2)A and S6(2)B Fig in S1 File). Compared with that of M0 treatment, the average root diameter of leymus chinensis in two batches are 0.20 mm and 0.19 mm promoted 10.77% (P<0.05) and 14.91% (P<0.05), respectively. However, increasing the amount of modifier has no significant effect on the average diameter of root system. The total root surface area of leymus chinensis with treatment M20 in two batches (15.86 $cm^2$ and 20.77 $cm^2$) are higher 28.00% (P<0.05) and 33.36% (P<0.05) than that of M0 treatment, respectively. Nonetheless, water-soluble fertilizer does not significantly increase the total root surface area of the M20 treatment. It can be seen from S6(4)A and S6(4)B Fig in S1 File the total root volume of leymus chinensis with treatment M0 in two batches are 71.67 $cm^3$ and 99.67 $cm^3$ which lower 23.56% (P<0.05) and 22.54% (P<0.05) than that of M20 treatment, respectively. In this study, it is found that all root growth indexes appear a trend of first increasing and then decreasing after adding modifier, and all indexes generally reach the maximum value when the amount of modifier is less than 20 g/kg, which is the same as the change trend of root growth of alfalfa.

**Root enzyme activity and rhizosphere soil enzyme activity.**   Fig 4 depicts the effects of modifier and water-soluble fertilizer on the dehydrogenase activity of leymus chinensis roots. The activity of roots dehydrogenase of leymus chinensis with M0 treatment in first planting is remarkably enhanced 10.15% (P<0.05) than that of treatment M0, is further increase by 20.11% adding water-soluble fertilizer. However,the activity of root dehydrogenase is not obviously increased by increasing the amount of modifier. The activity of root dehydrogenase of leymus chinensis with M20 is the optimum value that improve 9.04% (P<0.05) than that of M0 treatment in second planting.

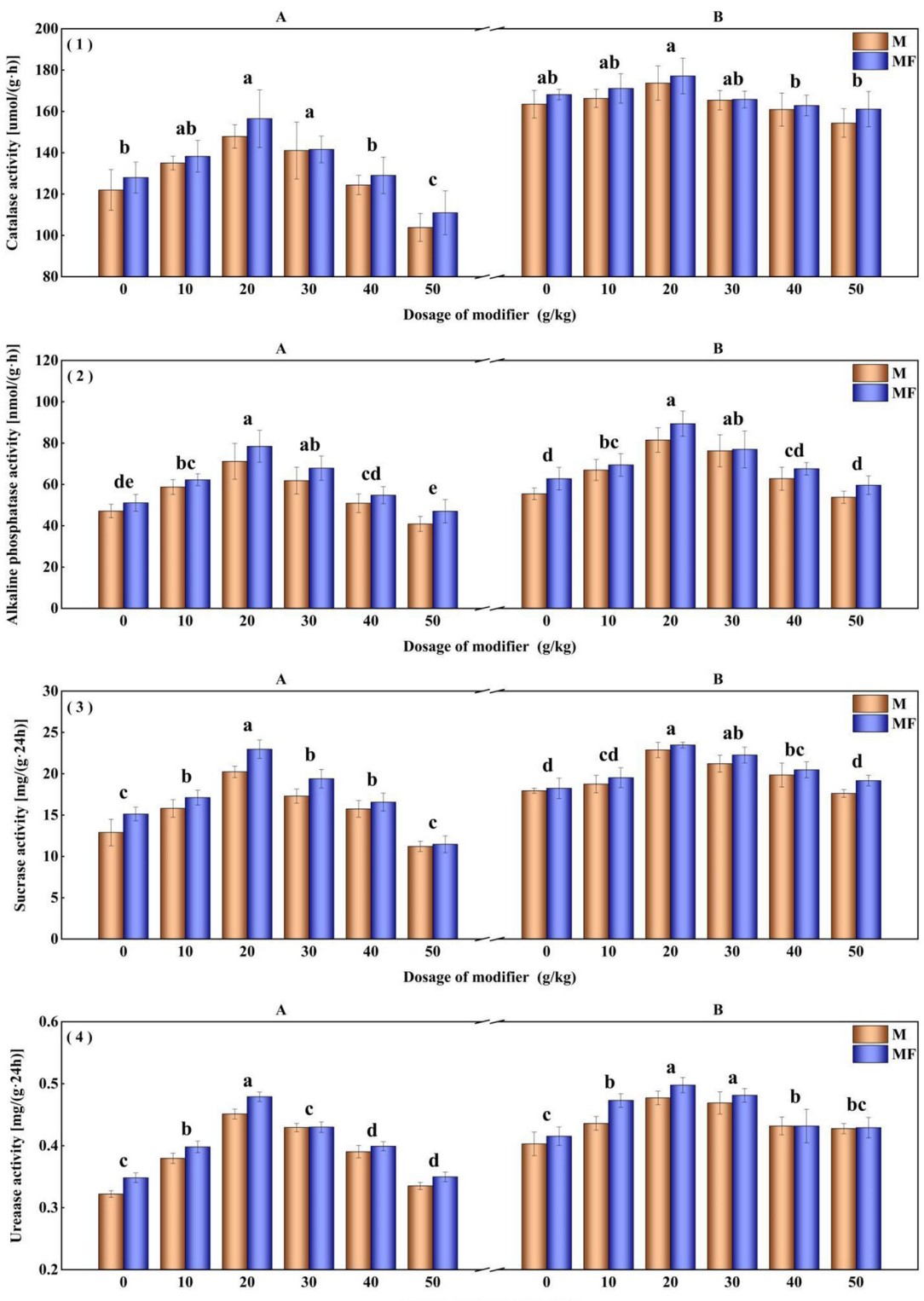

**Fig 3. Effects of modifier and water-soluble fertilizer on rhizosphere soil enzyme activities of alfalfa.** (1) Catalase, (2) Alkaline phosphatase, (3) Sucrase and (4) Urease.

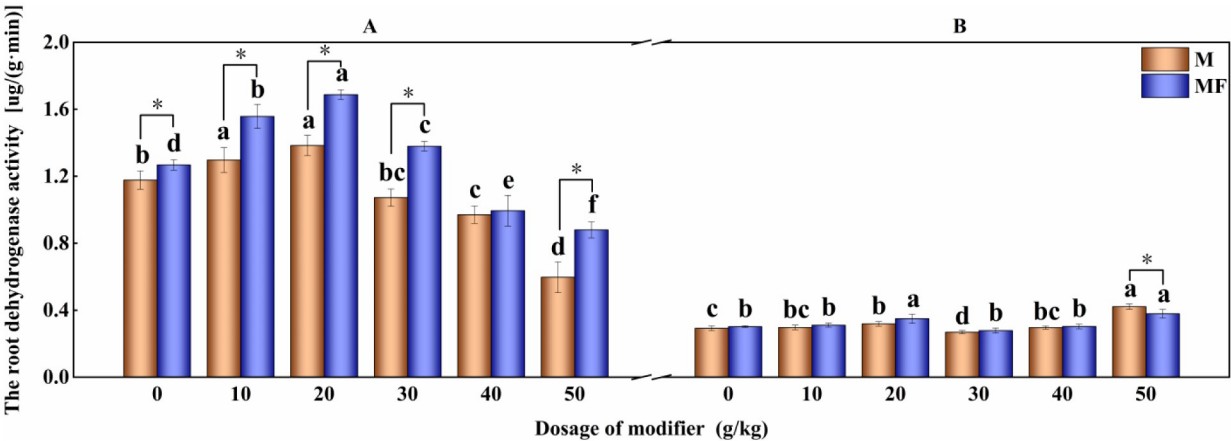

**Fig 4. Effects of modifier and water-soluble fertilizer on dehydrogenase activity of leymus chinensis roots.**

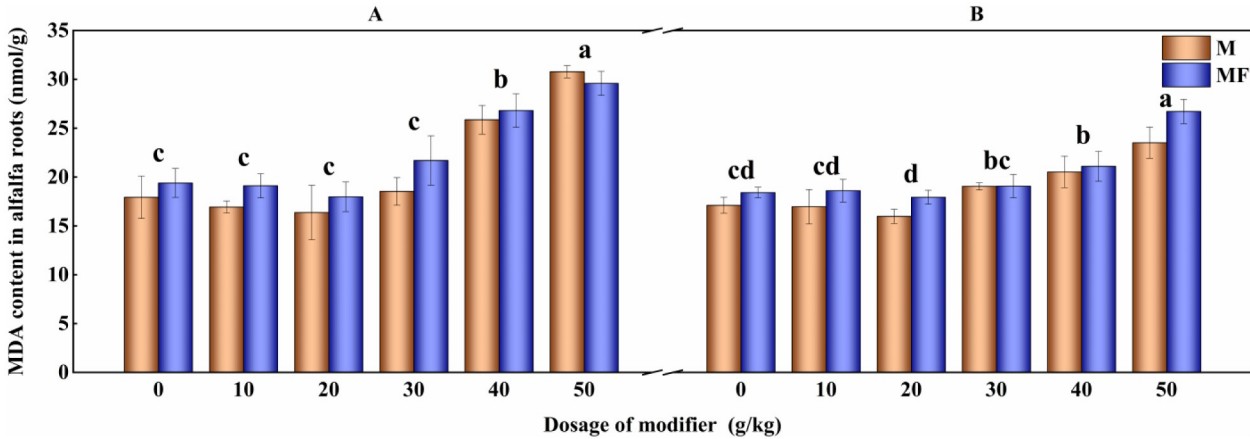

**Fig 5. Effect of modifier and water-soluble fertilizer on MDA content in leymus chinensis roots.**

Fig 5 exhibits the effects of modifier and water-soluble fertilizers on the malondialdehyde content of leymus chinensis roots. The content of malondialdehyde of the roots in two batches get the lowest value when the amounts of modifier is 20 g/kg, The content of malondialdehyde in the roots of leymus chinensis has a more pronounced promoting which the amount of modifier is more than 20 g/kg.

After the application of modifier, the change trend of soil enzyme activity in leymus chinensis is similar to that of alfalfa. The activities of alkaline phosphatase, sucrase and urease in rhizosphere soil with treatment M20 in second planting are promoted 44.52% ($p<0.05$), 17.30% ($p<0.05$) and 10.11% ($p<0.05$) than that of those treated with M0, respectively (Fig 6(1–4)).

## Results of two factor analysis variance

Two-factor ANOVA is a common statistical analysis used to compare whether there is an interaction effect between two factors [45]. The analysis of variance will examine the effects of soil modifiers and water-soluble fertilizers on the root system and rhizosphere soil enzyme

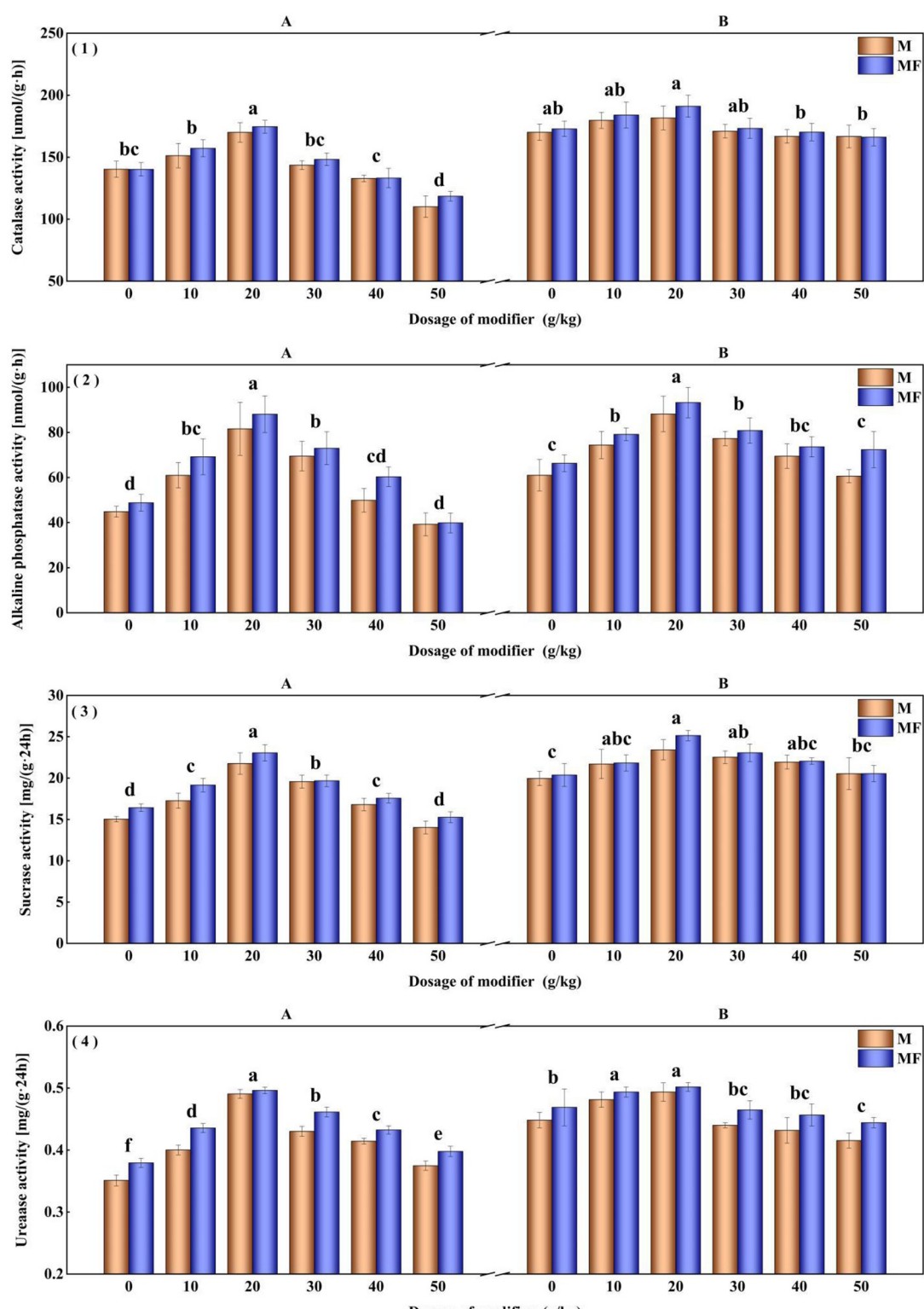

**Fig 6. Effects of modifier and water-soluble fertilizer on rhizosphere soil enzyme activities of leymus chinensis.** (1) Catalase, (2) Alkaline phosphatase, (3) Sucrase and (4) Urease.

activities of the two forages. The study aims to identify and compare the specific impacts of modifiers and water-soluble fertilizers on the soil at different time points. The two-factor experiments are conducted to analyze the changes in forage root indexes following the application of soil modifiers and water-soluble fertilizers. The results of the two-factor analysis of variance indicate that the soil modifier significantly influences the root indexes of both forage types. On the other hand, the water-soluble fertilizer prominently affects the total root length, total root surface area, total root volume, root dehydrogenase activity, root malondialdehyde content, and other related indexes of the two forage batches, as shown from Tables 1 to 4.

In F test, the F value of root indexes of the two batches applied of the modifier are generally higher than that of water-soluble fertilizer, indicating that the modifier has a greater effect on the roots of the two forages.

The F value of each treatment with the first planting applied modifier is greater than that of the second planting. The reason may be that a part of the modifier in the soil is absorbed by

**Table 1. Two-factor analysis of variance of the effects of modifier and water-soluble fertilizers on the roots of alfalfa.**

| First batch | | Survival number | Root biomass | Total root length | Average root diameter | Total root surface area | Total root volume | DHA | MDA |
|---|---|---|---|---|---|---|---|---|---|
| Modifier | F | 114.320 | 330.854 | 1389.080 | 30.067 | 404.009 | 228.624 | 77.155 | 356.400 |
| | P | 0.000 | 0.000 | 0.000 | 0.000 | 0.000 | 0.000 | 0.000 | 0.000 |
| Water-soluble fertilizer | F | 5.536 | 33.395 | 91.192 | 3.473 | 16.172 | 8.495 | 18.668 | 58.717 |
| | P | 0.027 | 0.000 | 0.000 | 0.075 | 0.000 | 0.008 | 0.000 | 0.000 |
| Modifier × Water-soluble fertilizer | F | 0.329 | 8.050 | 5.802 | 0.765 | 1.847 | 0.859 | 1.327 | 5.759 |
| | P | 0.890 | 0.000 | 0.001 | 0.584 | 0.142 | 0.523 | 0.287 | 0.001 |
| Second batch | | Survival number | Root biomass | Total root length | Average root diameter | Total root surface area | Total root volume | DHA | MDA |
| Modifier | F | 35.497 | 25.109 | 181.199 | 14.878 | 70.910 | 54.659 | 149.821 | 34.443 |
| | P | 0.000 | 0.000 | 0.000 | 0.000 | 0.000 | 0.000 | 0.000 | 0.000 |
| Water-soluble fertilizer | F | 6.125 | 5.788 | 9.985 | 5.200 | 9.797 | 18.073 | 9.938 | 99.647 |
| | P | 0.021 | 0.024 | 0.000 | 0.032 | 0.005 | 0.000 | 0.004 | 0.000 |
| Modifier × Water-soluble fertilizer | F | 0.525 | 0.118 | 1.736 | 0.941 | 0.349 | 2.405 | 4.877 | 0.951 |
| | P | 0.755 | 0.987 | 0.165 | 0.472 | 0.878 | 0.067 | 0.003 | 0.467 |

**Table 2. Two-factor analysis of variance of the effects of modifier and water-soluble fertilizers on the rhizosphere soil enzyme activities of alfalfa.**

| Alfalfa (First batch) | | CAT | ALP | SC | URE |
|---|---|---|---|---|---|
| Modifier | F | 18.339 | 26.925 | 72.028 | 245.592 |
| | P | 0.000 | 0.000 | 0.000 | 0.000 |
| Water-soluble fertilizer | F | 2.910 | 8.321 | 21.529 | 38.162 |
| | P | 0.101 | 0.008 | 0.000 | 0.000 |
| Modifier × Water-soluble fertilizer | F | 0.162 | 0.126 | 1.284 | 2.650 |
| | P | 0.974 | 0.985 | 0.303 | 0.048 |
| Alfalfa (Second batch) | | CAT | ALP | SC | URE |
| Modifier | F | 5.393 | 22.789 | 26.829 | 23.015 |
| | P | 0.002 | 0.000 | 0.000 | 0.000 |
| Water-soluble fertilizer | F | 3.030 | 6.793 | 6.702 | 7.502 |
| | P | 0.095 | 0.015 | 0.016 | 0.011 |
| Modifier × Water-soluble fertilizer | F | 0.196 | 0.377 | 0.319 | 1.202 |
| | P | 0.961 | 0.859 | 0.896 | 0.338 |

**Table 3. Two-factor analysis of variance of the effects of modifier and water-soluble fertilizers on the root system of Leymus chinensis.**

| First batch | | Survival number | Root biomass | Total root length | Average root diameter | Total root surface area | Total root volume | DHA | MDA |
|---|---|---|---|---|---|---|---|---|---|
| Modifier | F | 56.518 | 556.223 | 361.024 | 29.947 | 278.396 | 123.731 | 136.852 | 58.610 |
| | P | 0.000 | 0.000 | 0.000 | 0.000 | 0.000 | 0.000 | 0.000 | 0.000 |
| Water-soluble fertilizer | F | 0.680 | 56.542 | 4.097 | 13.820 | 20.493 | 21.902 | 107.987 | 5.854 |
| | P | 0.418 | 0.000 | 0.054 | 0.001 | 0.000 | 0.000 | 0.000 | 0.023 |
| Modifier × Water-soluble fertilizer | F | 1.222 | 1.775 | 0.186 | 0.389 | 0.693 | 8.367 | 6.032 | 1.121 |
| | P | 0.329 | 0.156 | 0.965 | 0.851 | 0.633 | 0.000 | 0.001 | 0.376 |
| Second batch | | Survival number | Root biomass | Total root length | Average root diameter | Total root surface area | Total root volume | DHA | MDA |
| Modifier | F | 8.966 | 11.993 | 123.178 | 7.863 | 107.749 | 49.574 | 46.985 | 38.816 |
| | P | 0.000 | 0.000 | 0.000 | 0.000 | 0.000 | 0.000 | 0.000 | 0.000 |
| Water-soluble fertilizer | F | 0.697 | 0.380 | 2.833 | 1.465 | 9.331 | 10.529 | 0.773 | 13.342 |
| | P | 0.412 | 0.543 | 0.105 | 0.238 | 0.005 | 0.003 | 0.388 | 0.001 |
| Modifier × Water-soluble fertilizer | F | 0.102 | 0.050 | 0.727 | 1.210 | 1.657 | 0.456 | 3.652 | 1.312 |
| | P | 0.991 | 0.998 | 0.610 | 0.335 | 0.184 | 0.805 | 0.013 | 0.292 |

**Table 4. Two-factor analysis of variance of the effects of modifier and water-soluble fertilizers on the rhizosphere soil enzyme activities of leymus chinensis.**

| Alfalfa (First batch) | | CAT | ALP | SC | URE |
|---|---|---|---|---|---|
| Modifier | F | 55.030 | 39.304 | 74.639 | 228.383 |
| | P | 0.000 | 0.000 | 0.000 | 0.000 |
| Water-soluble fertilizer | F | 3.394 | 6.515 | 17.591 | 93.860 |
| | P | 0.078 | 0.017 | 0.000 | 0.000 |
| Modifier × Water-soluble fertilizer | F | 0.391 | 0.448 | 0.893 | 3.208 |
| | P | 0.850 | 0.810 | 0.501 | 0.023 |
| Alfalfa (Second batch) | | CAT | ALP | SC | URE |
| Modifier | F | 6.477 | 18.238 | 10.263 | 17.832 |
| | P | 0.001 | 0.000 | 0.000 | 0.000 |
| Water-soluble fertilizer | F | 2.036 | 9.476 | 1.616 | 15.671 |
| | P | 0.167 | 0.005 | 0.216 | 0.001 |
| Modifier × Water-soluble fertilizer | F | 0.283 | 0.430 | 0.467 | 0.421 |
| | P | 0.918 | 0.823 | 0.797 | 0.829 |

the forage after the first planting, thus reducing the effect of the modifier on the forages. In two batches, the modifier has remarkable impact on enzyme activities in rhizosphere soil of herbage because the F value of various enzyme activities in rhizosphere soil of herbage after the application of the modifier is generally higher than that of water-soluble fertilizer.

## RDA analysis

RDA analysis can depict the samples and environmental factors on the same two-dimensional ordination plot, from which the relationship between sample distribution and environmental factors can be visualized [46]. The redundancy analysis of the indexes on two kind of forages and the four kind of enzyme activity of rhizosphere soil are carried out. The relationship between the first axis and the second axis of characteristics root growth indexes and enzyme activity in rhizosphere soil in the first planting alfalfa is analyzed by redundancy analysis, and

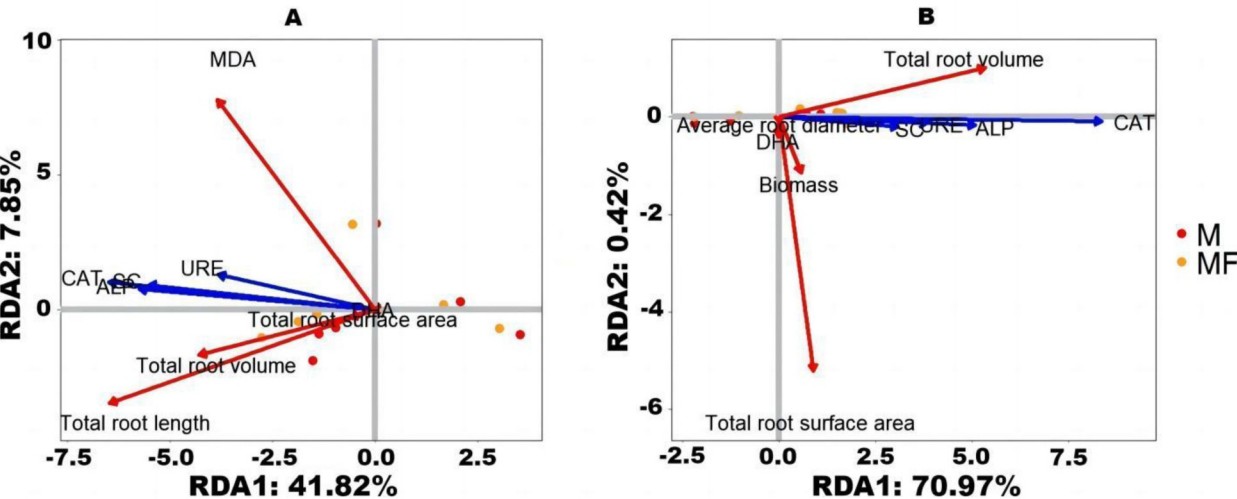

**Fig 7. Redundancy analysis results between root index of alfalfa and rhizosphere soil enzyme activity.**

the cumulative explanatory variables are 48.56% and 1.73% (Fig 7A), respectively. That of in second planting are 68.06% and 2.18% (Fig 7B), respectively. The enzyme activity in rhizosphere soil of two forages of alfalfa had a play an important role on all the root indexes analyzed, and is a crucial factor affecting the root indexes. The analysis results illustrate that there is a positive correlation between root index and rhizosphere soil enzyme activity, CAT and ALP has a higher correlation with sample distribution, while URE and SC has a lower correlation.

Fig 8A and 8B display the relationship between the first axis and the second axis of characteristics root growth indexes and enzyme activity in rhizosphere soil in the first planting leymus chinensis is measured by redundancy analysis. The cumulative explanatory variables are 828.68% and 2.72% (Fig 8A) and 746.38% and 0.77% (Fig 8B), respectively. The enzyme activity in rhizosphere soil of two forages of alfalfa has a certain promoting effect on most of the

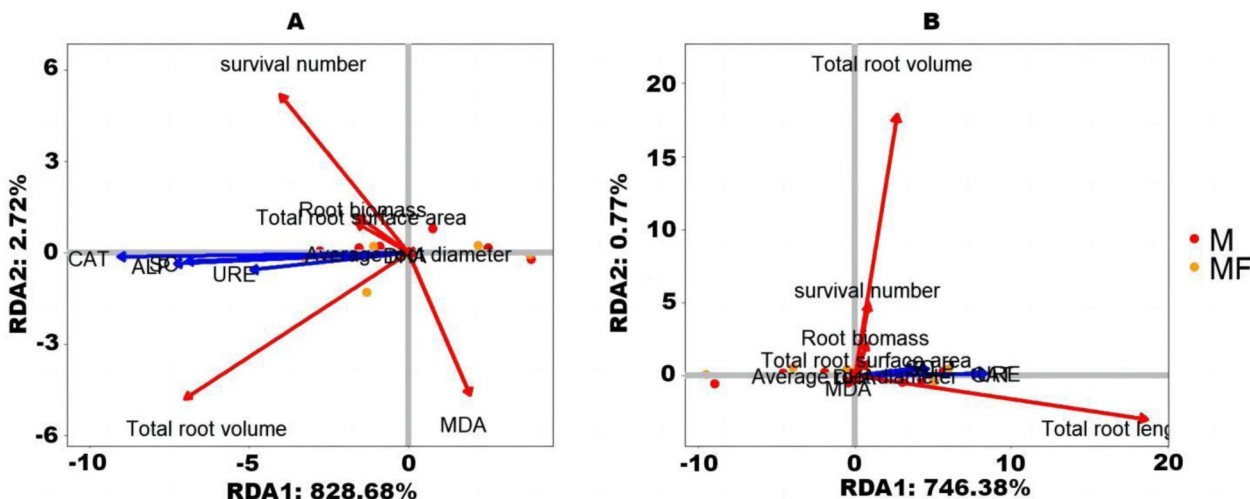

**Fig 8. Redundancy analysis results between root index of leymus chinensis and rhizosphere soil enzyme activity.**

root indexes. However, it does not promote on the content of malondialdehyde. Therefore, it can be seen that the enzyme activity in rhizosphere soil also affected the root growth of leveus chinensis, but the effect is less than that of alfalfa.

## Discussion

### Effects of modifier and water-soluble fertilizer on alfalfa

The present study consolidates previous test methods and determines the optimal modifier dosage, ranging from 10 g/kg to 50 g/kg. The study reveals that an excessive amount of modifier adversely affects the soil, leading to failure of alfalfa germination, hindered growth, and reduced survival rate. The optimal modifier amount is 20 g/kg, and it exhibits a stronger influence when applied with water-soluble fertilizer. However, if the modifier amount exceeds 40 g/kg, an excessive amount of black material will be present on the water surface after adding water. Over time, this material will solidify on the soil surface, resulting in hardened soil and making it challenging for seeds to emerge. This phenomenon may be the primary cause for the decreased survival of alfalfa when an excessive amount of modifier is used. Because hydrochar and fine sand do not float on the water surface in this experiment, and fungus chaff is yellow. It is assumed that the black substance floating on the water surface is humic acid. Humic acid, like inorganic acid, reacts with alkali to form a water-soluble humic acid salt [47]. It is evident that an excessive amount of humic acid should be avoided, as supported by the findings of this study. Overdosing the modifiers not only hinders soil absorption but also wastes resources. In addition, the excess surface modifiers carry salt, which is easily transferred with water. When water is added, the excessive surface modifiers release salt into the soil, resulting in a substantial increase in the soil's water-soluble salt content. This influx of salt has a negative impact on the soil. If the modifier amount exceeds 20 g/kg, further increasing the dosage will lead to a significant rise in water-soluble salt content and soil alkalinity.

The root is the tissue which plants extract nutrients and water from soil, and it has a significant impact on plant growth and crop yield [48, 49]. On the one hand, plants absorb water and nutrients through their roots, and the root surface area can directly reflect absorption area by the root. On the other hand, the roots can ensure the normal growth of the above-ground part of the plant [50]. Furthermore, root channels are an important component of soil macropores [24]. The average diameter of roots provides information about the relationship between soil pore size and root penetration. Therefore, root growth is beneficial to improving soil porosity which is critical to the improvement of saline-alkali soil. In this experiment, the modifier contains biochar, mushroom bran, humic acid, and other ingredients. Biochar contains a small amount of available phosphorus, which can be absorbed by the soil [51]. It can also adsorb nutrients in the soil [52], improve soil structure [34] and promoting alfalfa root growth. The fungus chaff is loose and porous which can increase soil porosity and benefit the growth and development of alfalfa root. Humic acid can increase soil nutrient content and improve soil physical and chemical properties [53–55], promoting the root growth of alfalfa. In this experiment, when the amount of modifier is 10 g/kg, most root indexes are improved significantly. When the modifier concentration is 20 g/kg, all indexes reach their maximum value. Roots absorb and accumulate salt from the soil, reducing salinization [56]. Simultaneously, higher root biomass mean that more roots are remained in the soil after crop harvest, and are decomposed to increase organic matter and inorganic nutrients in the soil, improving soil fertility [57], This is also one of the ways to improve saline-alkali soil. The researcher found that modified biochar could promote the growth of miscanthus sinensis on saline-alkali soil after 92 days of pot experiment. When the amount of modified biochar was 2%—2.5%, the root biomass of miscanthus sinensis was significantly increased [58]. In this experiment, the planting

duration for alfalfa was 50 days. When a modifier amount of 10 g/kg was used, there was a significant enhancement in root biomass. Moreover, the cost of the modifiers used in this experiment was lower compared to the cost of applying modified biochar alone. These results highlight the advantages of the modifier employed in this study. Additionally, the application of water-soluble fertilizer substantially increased both root length and root biomass in the initial alfalfa planting. The modifier used in this study shows effective improvement in saline-alkali soil. However, it lacks essential components such as glucose, organic acids, as well as calcium and magnesium ions that are necessary for plant growth. The application of water-soluble fertilizer helps to compensate for these deficiencies by providing the required nutrients to support plant growth [59, 60]. As a result, combination of the modifier and water-soluble fertilizer application is more conducive to improvement of saline-alkali soil and crop growth on saline-alkali soil. The water-soluble fertilizer used in this experiment is composed of an acid-mixed solution left over from the hydrocar of straw, calcium nitrate tetrahydrate, magnesium nitrate hexahydrate, and other ingredients. $Ca^{2+}$ can reduce the absorption of $Na^+$ by plants, maintain a low $Na^+/K^+$ ratio in cells, and improve the tolerance of plants to salt [61, 62]. $Mg^{2+}$ is a component of chlorophyll, which has an important impact on plant photosynthesis [63]. At the same time, as the most abundant bivalent free cation in cells, $Mg^{2+}$ is crucial for plant root growth [64]. Organic acids can improve physical and chemical properties of soil, decrease content of $Na^+$ in saline-alkali soil, increase soil desalination rate, and effectively improve the improvement effect of saline-alkali soil [65, 66]. In addition, it is discovered that water-soluble fertilizer further promote the growth of alfalfa leaves. The water-soluble fertilizer is made from the liquid waste left over from the hydrothermal reaction process of biomass, which can not only improve the saline-alkali soil and promote the growth of crop root, but also further prevent the waste of resource. Comparison of various indexes of alfalfa root growth in the two batches experiments, the effect of the modifier on the root growth of alfalfa in the first planting is better than that of the second planting, which indicates that alfalfa in the first planting absorb more nutrients in the modifier. However, the indexes in the second planting have a greater improvement compared with the first planting when the amounts of modifier reach 40 g/kg or above. The results indicate that the excess nutrients in the modifier are consumed in the first crop experiment, and the remaining nutrients in the second crop soil are more suitable for alfalfa growth. Abiala find that alfalfa can improve soil environment by reducing soil salinity and nitrogen fixation via rhizobium microorganisms [67]. This is also the reason why the root indexes of alfalfa in the second planting are higher than that of the first planting.

Root dehydrogenase is closely related to the growth and development of crop roots, and the level of its activity can reflect the strength of roots and is an indicator representing root vitality [68], which can be used to determine the root growth status of plants [69]. This experiment demonstrates that applying modifier could significantly increase the activity of dehydrogenase in the roots of two batches alfalfa, which could be related to the composition of the modifier. Nitrogen application to the soil has been shown in studies to increase crop root dehydrogenase activity [70]. The spent mushroom substrate used in this experiment contains a high level of nitrogen [71]. Humic acid is also helpful in converting total nitrogen to available nitrogen [72]. Therefore, the dehydrogenase activity of alfalfa roots is boosted by the addition of modifier. Dehydrogenase activity is not only related to growth status, but also to growth space. The higher the number of growth per unit area, the lower the dehydrogenase activity, and if the lower the survival rate, the relative increase in dehydrogenase activity. This could be the primary reason that the dehydrogenase activity of the first planting of alfalfa is higher than that of the second planting of alfalfa which is also reason the dehydrogenase activity of the second plating of alfalfa gradually increases when the modifier concentration are 40 g/kg and 50 g/kg.

The level of malondialdehyde content in alfalfa roots can reflect the degree of damage to the cell membrane [73], which destroyed a lot of biological functional molecules in root cells such as proteins and enzymes. To reduce malondialdehyde accumulation is an effective way to protect root growth. The content of malondialdehyde in the two batches of alfalfa roots is significantly reduce with 20 g/kg modifier in this experiment indicating that the addition of the modifier has a certain direct effect on the accumulation of malondialdehyde in alfalfa roots. Li et al. [74] discovered that the content of malondialdehyde in plants increased as the growth period progressed.

In the first planting of alfalfa in the field micro-area experiment, it is observed form that the growth rate of alfalfa is conspicuously slower with modifier amounts of 40 g/kg and 50 g/kg compared to the other treatments. As a result, the content of malondialdehyde, an indicator of oxidative stress, is lower in these treatments. However, in the second planting of the field micro-area experiment, from that when the modifier amounts exceeded 40 g/kg, the growth rate of alfalfa does not exhibit a significant decrease. The negligible disparity in the growth period of various treatments on alfalfa during the second planting may account for this observation. Notably, the root growth index indicates that the adverse effects of an excessive modifier on root growth in the second planting are substantially less pronounced compared to those observed during the initial planting. This further validates the accuracy of this perspective.

Soil enzyme activity is an important indicator of soil environment under saline-alkali stress [75]. Soil catalase primarily decomposes hydrogen peroxide in soil and mitigates the harm of excessive accumulation hydrogen peroxide in soil to plant roots. Soil phosphatase, an essential factor in determining the accessibility of soil phosphorus, serves as an indicator for assessing the magnitude and direction of soil phosphorus biotransformation. Among the various types of phosphatases, alkaline phosphatase is capable of breaking down phosphorus compounds within the soil, thereby enhancing their solubility [76] and decompose hydrolyze organic phosphates to inorganic phosphates to enhance the supply of soil phosphorus [77]. In terrestrial ecosystems, alkaline phosphatase is mainly produced by microorganisms [78]. Soil sucrase is also called soil sucrase invertase. Therefore, the sucrase often are measured in soil due to the most active in acidic media which is also acidic invertase. Soil urease is a hydrolase that decomposes organic matter containing nitrogen. It is commonly found in all fungi and serves as a direct source of nitrogen nutrition for plants. It can convert organic nitrogen in soil into crop-available nitrogen [79] and is essential for soil material transformation and energy metabolism.

Biochar contains a high concentration of nutrients, which can improve soil fertility [80], promoting the growth of soil microorganisms and enhancing enzyme activity. The rich crude protein, crude fat, and other components in spent mushroom substrate provide abundant nutrients for soil microorganism growth and reproduction, and microorganism metabolites also increase the activity of soil enzymes.

In this experiment, the modifier includes hydrothermal biochar. Being rich in nutrients, it has the potential to enhance soil enzyme activity and stimulate root growth. The roots play a vital role in nutrient extraction from the soil and release compounds into the rhizosphere, thereby altering the properties of the rhizosphere soil and subsequently improving its enzyme activity. Previous studies have demonstrated a significant correlation between the nutrient dynamics of the rhizosphere soil and the quality and quantity of root exudes [81, 82]. Henceforth, it is evident that root exudates play a significant role in influencing the enzyme activity of the rhizosphere soil, aligning with the findings of this study. Notably, water-soluble fertilizer, typically applied through foliar spraying, has a direct influence on root development and subsequently affects the enzyme activity within the rhizosphere soil of crops. As observed in

this experiment, the application of water-soluble fertilizer resulted in improved enzyme activity across various rhizosphere soils. It is worth emphasizing that soil enzymes play a crucial role in supplying the essential nutrients necessary for optimal plant growth [83].

## Effects of modifier and water-soluble fertilizer on leymus chinensis

Leymus chinensis has a thick and long underground rhizome, which can survive and grow in soil containing 600 mM NaCl and 175 mM $Na_2CO_3$ [84]. The high nutritional value and good taste also make leymus chinensis the most promising grass species for ecological restoration of saline-alkali soil and animal husbandry support [85–87]. The results of this experiment demonstrate that, within the context of a saline-alkali environment, leymus chinensis exhibits a higher survival rate and greater root biomass compared to alfalfa. However, it is important to note that leymus chinensis seedlings do not possess the characteristic of rapid ground coverage, nor do they contribute to the reduction of soil moisture evaporation. As a consequence, the primary benefit of cultivating leymus chinensis in saline-alkali soil lies in the accelerated growth of its root system. The experimental findings reveal that the survival rates and root biomass of leymus chinensis display similar patterns to those of alfalfa following the introduction of modifiers and water-soluble fertilizers. The underlying reasons behind these changes, induced by the application of modifiers and water-soluble fertilizers, are fundamentally akin to those experienced by alfalfa. However, a notable distinction arises in the response to water-soluble fertilizers. While the first planting of alfalfa significantly increased its root biomass, no such substantial effect is observed for leymus chinensis. This suggests that the impact of water-soluble fertilizers on root growth in leymus chinensis is comparatively less pronounced when compared to alfalfa.

The results of the experiment indicate that as the quantity of modifier is increased, the trends observed in the root growth index of leymus chinensis closely resemble those of alfalfa. However, it is worth noting that the average diameter of the second planting of leymus chinensis does not surpass that of the first planting. Moreover, when the amount of modifier reaches 20 g/kg, the application of water-soluble fertilizer does not exhibit a significant increase in the root growth index of leymus chinensis. This suggests that the root growth of leymus chinensis is comparatively less influenced by water-soluble fertilizer in this particular scenario.

The experimental results display that with the increase of the amount of modifier, the change trend of various root growth indexes of leymus chinensis is basically the same as that of alfalfa, but the average root diameter of leymus chinensis in the second plating is not higher than that in the first planting, water-soluble fertilizer does not significantly improve the root growth indexes of leymus chinensis with 20 g/kg modifier. This further indicates that the root growth of leymus chinensis is less affected by water-soluble fertilizer.

In the second planting, the dehydrogenase activity of leymus chinensis increase with the increase of the amounts of modifier when the amount of modifier less than 30 g/kg. Over optimum amount led to the small survival amount and large growth space of single crop, which also indicate that excessive planting density of forage would have a negative impact on root vitality.

The results illustrate that applying modifier has no significant effect on reducing malondialdehyde content in leymus chinensis roots. The content of malondialdehyde begin to rise when the amount of modifier reach 30 g/kg or higher, and the peak at 50 g/kg, which is significantly higher than in other treatments.

The change trend of enzyme activity in the rhizosphere soil of leymus chinensis is similar to that of alfalfa, and the reason for the change ls essentially the same as that of alfalfa. However, under the same conditions, the enzyme activity of leymus chinensis in the rhizosphere soil is

higher than that of alfalfa. The main reason is that leymus chinensis had better root growth than alfalfa in the same environment, indicating that root growth has a significant effect on enzyme activity in the rhizosphere soil.

## Conclusions

1. The survival number, root biomass, and root growth index of forage species increase at first and then decrease with the increase of the amount of modifier from hydrothermal carbon source, reaching a maximum value at 20 g/kg modifier. And the application of medium quantity element water-soluble fertilizer could play a better effect. It can be seen that both modifier and water-soluble fertilizer promote the root growth of the two forage species and can better repair saline-alkali soil and increase crop yield. When the dosage of the modifier is less than 20 g/kg, the dehydrogenase activity of grass roots shows an increasing trend, and the malondialdehyde content show a decreasing trend. And the change is irregular after the dosage is more than 20 g/kg, which may be caused by the growth space and growth time.

2. The modifier and water-soluble fertilizer can effectively improve the rhizosphere soil enzyme activity of two forages species, but the amount should not be too much. When the modifier is applied at 20 g/kg, the alkaline phosphatase, sucrase and urease all reach the maximum value, and the effect of the modifier on the catalase activity is not obvious.

## Supporting information

**S1 File.**
(DOCX)

## Author Contributions

**Conceptualization:** Shengchen Zhao, Haibo Chang.

**Data curation:** Shengchen Zhao, Jingmin Yang.

**Formal analysis:** Shengchen Zhao, Dapeng Wang.

**Funding acquisition:** Haibo Chang.

**Methodology:** Yunhui Li, Wei Wang.

**Writing – original draft:** Shengchen Zhao, Jihong Wang.

**Writing – review & editing:** Dapeng Wang, Haibo Chang.

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
