## [Decision Letter · Decision Letter 0]

21 Nov 2023

PONE-D-23-28238The effect of modifier and a water-soluble fertilizer on two forages grown in saline-alkaline soilPLOS ONE

Dear Dr. Chang,

Thank you for submitting your manuscript to PLOS ONE. After careful consideration, we feel that it has merit but does not fully meet PLOS ONE’s publication criteria as it currently stands. Therefore, we invite you to submit a revised version of the manuscript that addresses the points raised during the review process.

We look forward to receiving your revised manuscript.

Kind regards,

Mehdi Rahimi, Ph.D.

Academic Editor

PLOS ONE

4. PLOS requires an ORCID iD for the corresponding author in Editorial Manager on papers submitted after December 6th, 2016. Please ensure that you have an ORCID iD and that it is validated in Editorial Manager. To do this, go to ‘Update my Information’ (in the upper left-hand corner of the main menu), and click on the Fetch/Validate link next to the ORCID field. This will take you to the ORCID site and allow you to create a new iD or authenticate a pre-existing iD in Editorial Manager. Please see the following video for instructions on linking an ORCID iD to your Editorial Manager account: https://www.youtube.com/watch?v=_xcclfuvtxQ.

Reviewers' comments:

Reviewer's Responses to Questions

**Comments to the Author**

1. Is the manuscript technically sound, and do the data support the conclusions?

Reviewer #1: Partly

2. Has the statistical analysis been performed appropriately and rigorously? 

Reviewer #1: Yes

3. Have the authors made all data underlying the findings in their manuscript fully available?

Reviewer #1: Yes

4. Is the manuscript presented in an intelligible fashion and written in standard English?

Reviewer #1: Yes

5. Review Comments to the Author

Reviewer #1: Introduction

This section lacks significant information about the deleterious impacts of sline soil on crop growth and development. In L41: before '' In order to ensure……'' The following paragraph is essential to be inserted in introduction section

• Salinity stress causes disturbance in plant physiological status (Abd El-Mageed et al. 2022; Lasheen et al. 2023) and imbalance nutrient absorption ((Mubarak et al. 2021; Salem et at. 2021)), causing week growth and low crop productivity and quality (Saudy et al. 2023; Shaaban et al. 2023)

Abd El-Mageed TA, Mekdad AAA, Rady MOA, Abdelbaky AS, Saudy HS, Shaaban A (2022) Physio-biochemical and agronomic changes of two sugar beet cultivars grown in saline soil as influenced by potassium fertilizer. J Soil Sci Plant Nutr. 22:3636-3654. https://doi.org/10.1007/s42729-022-00916-7

Lasheen FF, Hewidy M, Abdelhamid AN, Thabet RS, Abass MMM, Fahmy Asmaa A, Saudy HS, Hassan KM (2023) Exogenous application of humic acid mitigates salinity stress on pittosporum (Pittosporum tobira) plant by adjusting the osmolytes and nutrient homeostasis. Gesunde Pflanzen. https://doi.org/10.1007/s10343-023-00939-9

Mubarak M, Salem EMM, Kenawey MKM, Saudy HS (2021) Changes in calcareous soil activity, nutrient availability, and corn productivity due to the integrated effect of straw mulch and irrigation regimes. J Soil Sci Plant Nutr. 21:2020-2031. https://doi.org/10.1007/s42729-021-00498-w

Salem EMM, Kenawey MKM, Saudy HS, Mubarak M (2021) Soil mulching and deficit irrigation effect on sustainability of nutrients availability and uptake, and productivity of maize grown in calcareous soils. Comm. Soil Sci. Plant Anal. 52: 1745-1761. https://doi.org/10.1080/00103624.2021.1892733

Saudy HS, Salem EMM, Abd El–Momen WR (2023) Effect of potassium silicate and irrigation on grain nutrient uptake and water use efficiency of wheat under calcareous soils. Gesunde Pflanzen. 75: 647-654. https://doi.org/10.1007/s10343-022-00729-9

Shaaban A, Abd El-Mageed TA, Abd El-Momen WR, Saudy HS, Al-Elwany OAAI (2023) The integrated application of phosphorous and zinc affects the physiological status, yield and quality of canola grown in phosphorus-suffered deficiency saline soil. Gesunde Pflanzen. 75: 1813-1821. https://doi.org/10.1007/s10343-023-00843-2

• The study involved the use of rice straw as a source of preparing hydrochar. Accordingly the background in Introduction section should be supported by knowledge related to the agricultural wastes and recycling before taking about biochar as follow:

It has been documented that amending soils, especially that suffering from environmental stresses, has a favorable effect on different soil properties (Saudy et al. 2021a; Abd–Elrahman et al. 2022; Ali et al. 2023). Thus, soil productivity and crop quality were improved by exploiting the recycled plant residues (El-Metwally et al. 2022a; El-Metwally et al. 2022b; Saudy et al. 2022).

Saudy HS, El-Bially, MA, Ramadan KhA, Abo El–Nasr EKh, Abd El-Samad GA (2021a) Potentiality of soil mulch and sorghum extract to reduce the biotic stress of weeds with enhancing yield and nutrient uptake of maize crop. Gesunde Pflanzen. 73:555-564. https://doi.org/10.1007/s10343-021-00577-z

Abd–Elrahman ShH, Saudy HS, Abd El–Fattah DA, Hashem FA (2022) Effect of irrigation water and organic fertilizer on reducing nitrate accumulation and boosting lettuce productivity. J Soil Sci Plant Nutr. 22:2144-2155. https://doi.org/10.1007/s42729-022-00799-8

Ali IAA, Hassan Soheir E, Abdelhafez AA, Hewidy M, Nasser MA, Saudy HS, Hassan KM, Abou-Hadid AF (2023) Modifying the growing media and bio stimulants supply for healthy gerbera (Gerbera jamesonii) flowers Gesunde Pflanzen. https://doi.org/10.1007/s10343-023-00943-z

El-Metwally IM Geries L, Saudy HS (2022a) Interactive effect of soil mulching and irrigation regime on yield, irrigation water use efficiency and weeds of trickle–irrigated onion. Archiv. Agron. Soil Sci. 68:1103-1116. https://doi.org/10.1080/03650340.2020.1869723

El-Metwally IM, Saudy HS, Elewa TA (2022b) Natural plant by-products and mulching materials to suppress weeds and improve sugar beet (Beta vulgaris L.) yield and quality. J Soil Sci Plant Nutr. 22:5217-5230 https://doi.org/10.1007/s42729-022-00997-4

Saudy HS, El-Metwally IM, Sobieh ST, Abd-Alwahed SHA (2022) Mycorrhiza, charcoal, and rocket salad powder as eco-friendly methods for controlling broomrape weed in inter-planted faba bean with flax. J Soil Sci Plant Nutr. 22:5195-5206 https://doi.org/10.1007/s42729-022-00995-6

• L69 support the sentence "Biochar is a novel soil modifier method for saline-alkali soil that significantly improves soil nutrients and properties" by the following citations

(El-Bially et al. 2023; Saudy et al. 2021b)

El-Bially MA, El-Metwally IM, Saudy HS, Aisa KH, Abd El-Samad GA (2023) Mycorrhiza-inoculated biochar as an eco-friendly tool improves the broomrape control efficacy in two faba bean cultivars. Rhizosphere 26:100706 https://doi.org/10.1016/j.rhisph.2023.100706

Saudy HS, Hamed MF, El–Metwally IM, Ramadan KhA, Aisa KH (2021b) Assessing the effect of biochar or compost application as a spot placement on broomrape control in two cultivars of faba bean. J Soil Sci Plant Nutr. 21:1856-1866. https://doi.org/10.1007/s42729-021-00485-1

Materials and methods

This section should be fully restructured

• L113: the author mentioned The amount of soil modifier from 10 g/kg……..Also, the same sentence was written in L122. There is overlapping between talking about modifier and soluble fertilizer

• Concerning measuring indicators and method, why authors did not add the related information in M&M section. It is not favorable to be added as supporting information. Further, I can no reach this file (supporting information)

• What is the academic reference used in Statistical analysis

Discussion

• L329: support the citation no. [29] by the following:

Makhlouf et al. 2022; Ramadan et al. 2023; Saudy et al.2020)

Makhlouf BSI, Khalil SRA, Saudy HS (2022) Efficacy of humic acids and chitosan for enhancing yield and sugar quality of sugar beet under moderate and severe drought. J Soil Sci Plant Nutr. 22:1676-1691. https://doi.org/10.1007/s42729-022-00762-7

Ramadan KMA, El-Beltagi HS, Abd El-Mageed TAA, Saudy HS, Al-Otaibi HH, Mahmoud MAA (2023) The changes in various physio-biochemical parameters and yield traits of faba bean due to humic acid plus 6-benzylaminopurine application under deficit irrigation. Agron 13:1227. https://doi.org/10.3390/agronomy13051227

Saudy HS, Hamed MF, Abd El–Momen WR, Hussein H (2020) Nitrogen use rationalization and boosting wheat productivity by applying packages of humic, amino acids and microorganisms. Comm. Soil Sci. Plant Anal 51:1036-1047. https://doi.org/10.1080/00103624.2020.1744631

• L344: support the citation no. [33] by the following:

(Abou El-Enin et al. 2023; Elgala et al. 2022)

Abou El-Enin MM, Sheha AM, El-Serafy Rasha S, Ali OAM, Saudy HS, Shaaban A (2023) Foliage-sprayed nano-chitosan-loaded nitrogen boosts yield potentials, competitive ability, and profitability of intercropped maize-soybean. Inter J Plant Prod 17: 517-542. https://doi.org/10.1007/s42106-023-00253-4

Elgala AM, Abd-Elrahman ShH, Saudy HS, Nossier MI (2022) Exploiting Eichhornia crassipes shoots extract as a natural source of nutrients for producing healthy tomato plants. Gesunde Pflanzen. 74:457-465. https://doi.org/10.1007/s10343-022-00622-5

6. PLOS authors have the option to publish the peer review history of their article (what does this mean?). If published, this will include your full peer review and any attached files.

Reviewer #1: No

---

## [Author Response · Author response to Decision Letter 0]

7 Dec 2023

Manuscript ID: PONE-D-23-28238

TITLE: The effect of modifier and a water-soluble fertilizer on two forages grown in saline-alkaline soil

Thank you very much for returning to us the Reviewers’ comments and queries. We also thank the Reviewers for their time and effort in evaluating our work. Please find our response to the points raised by the Reviewers in blue; changes in the manuscript are highlighted in red. We hope the revised manuscript can be accepted for publication on PLoS One.

Reviewer 1 comments:

1. This section lacks significant information about the deleterious impacts of sline soil on crop growth and development. In L41: before '' In order to ensure……'' The following paragraph is essential to be inserted in introduction section

• Salinity stress causes disturbance in plant physiological status (Abd El-Mageed et al. 2022; Lasheen et al. 2023) and imbalance nutrient absorption ((Mubarak et al. 2021; Salem et at. 2021)), causing week growth and low crop productivity and quality (Saudy et al. 2023; Shaaban et al. 2023)

Thanks for your suggestion. I have revised the abstract in the manuscript.

Introduction revised in the manuscript:

Additionally, it causes nutrient deficiency and accumulation of salt in saline-alkali soil, leading to water pollution[6,7]. Salinity stress causes disturbance in plant physiological status[8,9] and imbalance nutrient absorption[10,11], causing week growth and low crop productivity and quality[12,13]. In order to ensure the availability of arable land and achieve consistent growth and stability in grain production, it is imperative to enhance and harness the potential of high-potential saline-alkali soil[14]. 

8. Abd El-Mageed TA, Mekdad AAA, Rady MOA, Abdelbaky AS, Saudy HS, Shaaban A. Physio-biochemical and agronomic changes of two sugar beet cultivars grown in saline soil as influenced by potassium fertilizer. J Soil Sci Plant Nutr. 2022; 22: 3636-3654. https://doi.org/10.1007/s42729-022-00916-7.

9. Lasheen FF, Hewidy M, Abdelhamid AN, Thabet RS, Abass MMM, Fahmy Asmaa A, Saudy HS, Hassan KM. Exogenous application of humic acid mitigates salinity stress on pittosporum (Pittosporum tobira) plant by adjusting the osmolytes and nutrient homeostasis. Gesunde Pflanzen. 2023. https://doi.org/10.1007/s10343-023-00939-9.

10. Mubarak M, Salem EMM, Kenawey MKM, Saudy HS. Changes in calcareous soil activity, nutrient availability, and corn productivity due to the integrated effect of straw mulch and irrigation regimes. J Soil Sci Plant Nutr. 2021; 21:2020-2031. https://doi.org/10.1007/s42729-021-00498-w.

11. Salem EMM, Kenawey MKM, Saudy HS, Mubarak M. Soil mulching and deficit irrigation effect on sustainability of nutrients availability and uptake, and productivity of maize grown in calcareous soils. Comm. Soil Sci. Plant Anal. 2021; 52: 1745-1761. https://doi.org/10.1080/00103624.2021.1892733.

12. Saudy HS, Salem EMM, Abd El-Momen WR (2023) Effect of potassium silicate and irrigation on grain nutrient uptake and water use efficiency of wheat under calcareous soils. Gesunde Pflanzen. 2023; 75: 647-654. 

https://doi.org/10.1007/s10343-022-00729-9.

13. Shaaban A, Abd El-Mageed TA, Abd El-Momen WR, Saudy HS, Al-Elwany OAAI. The integrated application of phosphorous and zinc affects the physiological status, yield and quality of canola grown in phosphorus-suffered deficiency saline soil. Gesunde Pflanzen. 2023; 75: 1813-1821. https://doi.org/10.1007/s10343-023-00843-2.

2. The study involved the use of rice straw as a source of preparing hydrochar. Accordingly the background in Introduction section should be supported by knowledge related to the agricultural wastes and recycling before taking about biochar as follow:

 It has been documented that amending soils, especially that suffering from environmental stresses, has a favorable effect on different soil properties (Saudy et al. 2021a; Abd–Elrahman et al. 2022; Ali et al. 2023). Thus, soil productivity and crop quality were improved by exploiting the recycled plant residues (El-Metwally et al. 2022a; El-Metwally et al. 2022b; Saudy et al. 2022).

Thanks for your suggestion. We have added the above in the Introduction section of the manuscript.

Introduction revised in the manuscript:

 It has been documented that amending soils, especially that suffering from environmental stresses, has a favorable effect on different soil properties[26-28]. Thus, soil productivity and crop quality were improved by exploiting the recycled plant residues[29-31]. Biochar is a novel soil modifier method for saline-alkali soil that significantly improves soil nutrients and properties.

26. Saudy HS, El-Bially, MA, Ramadan KhA, Abo El–Nasr EKh, Abd El-Samad GA. Potentiality of soil mulch and sorghum extract to reduce the biotic stress of weeds with enhancing yield and nutrient uptake of maize crop. Gesunde Pflanzen. 2021a; 73: 555-564. https://doi.org/10.1007/s10343-021-00577-z.

27. Abd–Elrahman ShH, Saudy HS, Abd El–Fattah DA, Hashem FA. Effect of irrigation water and organic fertilizer on reducing nitrate accumulation and boosting lettuce productivity. J Soil Sci Plant Nutr. 2022; 22: 2144-2155. https://doi.org/10.1007/s42729-022-00799-8.

28. Ali IAA, Hassan Soheir E, Abdelhafez AA, Hewidy M, Nasser MA, Saudy HS, Hassan KM, Abou-Hadid AF. Modifying the growing media and bio stimulants supply for healthy gerbera (Gerbera jamesonii) flowers Gesunde Pflanzen. 2023. https://doi.org/10.1007/s10343-023-00943-z.

29. El-Metwally IM Geries L, Saudy HS. Interactive effect of soil mulching and irrigation regime on yield, irrigation water use efficiency and weeds of trickle–irrigated onion. Archiv. Agron. Soil Sci. 2022a; 68:1103-1116. https://doi.org/10.1080/03650340.2020.1869723.

30. El-Metwally IM, Saudy HS, Elewa TA. Natural plant by-products and mulching materials to suppress weeds and improve sugar beet (Beta vulgaris L.) yield and quality. J Soil Sci Plant Nutr. 2022b; 22:5217-5230 https://doi.org/10.1007/s42729-022-00997-4.

31. Saudy HS, El-Metwally IM, Sobieh ST, Abd-Alwahed SHA. Mycorrhiza, charcoal, and rocket salad powder as eco-friendly methods for controlling broomrape weed in inter-planted faba bean with flax. J Soil Sci Plant Nutr. 2022; 22: 5195-5206 https://doi.org/10.1007/s42729-022-00995-6.

3. L69 support the sentence "Biochar is a novel soil modifier method for saline-alkali soil that significantly improves soil nutrients and properties" by the following citations

(El-Bially et al. 2023; Saudy et al. 2021b)

Thanks for your suggestion. I have revised citations in the manuscript.

Introduction revised in the manuscript:

Biochar is a novel soil modifier method for saline-alkali soil that significantly improves soil nutrients and properties[32,33]. Several studies have shown that biochar has a greater improvement effect on saline-alkali soil compared to straw, attributed to its surface’s rich functional groups that can absorb and enhance nutrient holding capacity[6,34].

32. El-Bially MA, El-Metwally IM, Saudy HS, Aisa KH, Abd El-Samad GA. Mycorrhiza-inoculated biochar as an eco-friendly tool improves the broomrape control efficacy in two faba bean cultivars. Rhizosphere 2023; 26: 100706 https://doi.org/10.1016/j.rhisph.2023.100706.

33. Saudy HS, Hamed MF, El–Metwally IM, Ramadan KhA, Aisa KH. Assessing the effect of biochar or compost application as a spot placement on broomrape control in two cultivars of faba bean. J Soil Sci Plant Nutr. 2021b; 21: 1856-1866. https://doi.org/10.1007/s42729-021-00485-1.

4. L113: the author mentioned The amount of soil modifier from 10 g/kg……..Also, the same sentence was written in L122. There is overlapping between talking about modifier and soluble fertilizer

We have added conditions on line 129.

Materials and methods revised in the manuscript:

On basis of the amounts of modifiers were 10 g/kg - 50 g/kg (In conjunction with the application of water-soluble fertilizer), which were marked with MF10, MF20, MF30, MF40, and MF50. The blank experiment was performed under the same conditions with only water-soluble fertilizer without modifier that was marked MF0.

5.Concerning measuring indicators and method, why authors did not add the related information in M&M section. It is not favorable to be added as supporting information. Further, I can no reach this file (supporting information)

 We have added this section to the manuscript species.

6. What is the academic reference used in Statistical analysis

We have added academic reference in Statistical analysis.

Materials and methods revised in the manuscript:

Two-factor ANOVA is a common statistical analysis used to compare whether there is an interaction effect between two factors [42]. The analysis of variance will examine the effects of soil modifiers and water-soluble fertilizers on the root system and rhizosphere soil enzyme activities of the two forages. 

RDA analysis can depict the samples and environmental factors on the same two-dimensional ordination plot, from which the relationship between sample distribution and environmental factors can be visualized[43]. The redundancy analysis of the indexes on two kind of forages and the four kind of enzyme activity of rhizosphere soil are carried out. 

42. Zhou M, Wei P, Deng L. Research on the factorial effect of science and technology innovation (STI) policy mix using multifactor analysis of variance (ANOVA). Journal of Innovation & Knowledge. 2022; 7(4): 100249. https://doi.org/10.1016/j.jik.2022.100249.

43. Fu P, Zhang W, Yang K, Meng F. A novel spectral analysis method for distinguishing heavy metal stress of maize due to copper and lead: RDA and EMD-PSD. Ecotoxicology and environmental safety, 2020; 206: 111211. https://doi.org/10.1016/j.ecoenv.2020.111211.

7.L329: support the citation no. [29] by the following:

(Makhlouf et al. 2022; Ramadan et al. 2023; Saudy et al.2020)

 Done, on line 355.

Discussion revised in the manuscript:

Humic acid can increase soil nutrient content and improve soil physical and chemical properties[50-52], promoting the root growth of alfalfa. In this experiment, when the amount of modifier is 10 g/kg, most root indexes are improved significantly. 

50. Makhlouf BSI, Khalil SRA, Saudy HS. Efficacy of humic acids and chitosan for enhancing yield and sugar quality of sugar beet under moderate and severe drought. J Soil Sci Plant Nutr. 2022; 22: 1676-1691. https://doi.org/10.1007/s42729-022-00762-7.

51. Ramadan KMA, El-Beltagi HS, Abd El-Mageed TAA, Saudy HS, Al-Otaibi HH, Mahmoud MAA. The changes in various physio-biochemical parameters and yield traits of faba bean due to humic acid plus 6-benzylaminopurine application under deficit irrigation. Agron. 2023; 13: 1227. https://doi.org/10.3390/agronomy13051227.

52. Saudy HS, Hamed MF, Abd El-Momen WR, Hussein H. Nitrogen use rationalization and boosting wheat productivity by applying packages of humic, amino acids and microorganisms. Comm. Soil Sci. Plant Anal. 2020; 51:1036-1047. https://doi.org/10.1080/00103624.2020.1744631.

8.L344: support the citation no. [33] by the following:

(Abou El-Enin et al. 2023; Elgala et al. 2022)

 Done, on line 370.

Discussion revised in the manuscript:

The application of water-soluble fertilizer helps to compensate for these deficiencies by providing the required nutrients to support plant growth[56,57]. As a result, combination of the modifier and water-soluble fertilizer application is more conducive to improvement of saline-alkali soil and crop growth on saline-alkali soil. 

56. Abou El-Enin MM, Sheha AM, El-Serafy Rasha S, Ali OAM, Saudy HS, Shaaban A. Foliage-sprayed nanochitosan-loaded nitrogen boosts yield potentials, competitive ability, and profitability of intercropped maize-soybean. Inter J Plant Prod. 2023; 17: 517-542. https://doi.org/10.1007/s42106-023-00253-4.

57. Elgala AM, Abd-Elrahman ShH, Saudy HS, Nossier MI. Exploiting Eichhornia crassipes shoots extract as a natural source of nutrients for producing healthy tomato plants. Gesunde Pflanzen. 2022; 74:457-465. 

https://doi.org/10.1007/s10343-022-00622-5.

Best regards

Haibo Chang

---

## [Decision Letter · Decision Letter 1]

26 Dec 2023

PONE-D-23-28238R1The effect of modifier and a water-soluble fertilizer on two forages grown in saline-alkaline soilPLOS ONE

Dear Dr. Chang,

Thank you for submitting your manuscript to PLOS ONE. After careful consideration, we feel that it has merit but does not fully meet PLOS ONE’s publication criteria as it currently stands. Therefore, we invite you to submit a revised version of the manuscript that addresses the points raised during the review process.

We look forward to receiving your revised manuscript.

Kind regards,

Mehdi Rahimi, Ph.D.

Academic Editor

PLOS ONE

Journal Requirements:

Reviewers' comments:

Reviewer's Responses to Questions

**Comments to the Author**

1. If the authors have adequately addressed your comments raised in a previous round of review and you feel that this manuscript is now acceptable for publication, you may indicate that here to bypass the “Comments to the Author” section, enter your conflict of interest statement in the “Confidential to Editor” section, and submit your "Accept" recommendation.

Reviewer #1: All comments have been addressed

Reviewer #2: All comments have been addressed

2. Is the manuscript technically sound, and do the data support the conclusions?

Reviewer #1: Yes

Reviewer #2: Yes

3. Has the statistical analysis been performed appropriately and rigorously? 

Reviewer #1: Yes

Reviewer #2: Yes

4. Have the authors made all data underlying the findings in their manuscript fully available?

Reviewer #1: Yes

Reviewer #2: Yes

5. Is the manuscript presented in an intelligible fashion and written in standard English?

Reviewer #1: Yes

Reviewer #2: Yes

6. Review Comments to the Author

Reviewer #1: The authors addressed the required corrections.

There no comments for the author, including concerns about dual publication, research ethics, or publication ethics.

Reviewer #2: Though most of the suggestions have been addressed but the introduction need further refinement with clear objectives and hypothesis.

7. PLOS authors have the option to publish the peer review history of their article (what does this mean?). If published, this will include your full peer review and any attached files.

Reviewer #1: No

Reviewer #2: No

---

## [Author Response · Author response to Decision Letter 1]

27 Dec 2023

Manuscript ID: PONE-D-23-28238

TITLE: The effect of modifier and a water-soluble fertilizer on two forages grown in saline-alkaline soil

Thank you very much for returning to us the Reviewers’ comments and queries. We also thank the Reviewers for their time and effort in evaluating our work. Please find our response to the points raised by the Reviewers in blue; changes in the manuscript are highlighted in red. We hope the revised manuscript can be accepted for publication on PLoS One.

Reviewer 2 comments:

1. Though most of the suggestions have been addressed but the introduction need further refinement with clear objectives and hypothesis.

Thanks for your suggestion. I have revised the Introduction in the manuscript.

Introduction revised in the manuscript:

 Because of its porosity, biochar can improve soil physical structure[35], promote seed germination and crop growth[36], and improve soil enzyme activity and remove soil pollutants. Biochar itself is rich in carbon, and its ashes contains mineral elements needed for plant growth, which can improve soil permeability and increase soil carbon stock. In addition, the oxygen-containing functional groups on the surface of biochar, such as -COOH, -OH, etc., can react with the salt ions adsorbed in the soil colloid and adsorb Na+, which is the most representative of saline and alkaline soils, and thus improve the salinity and alkalinity of the soil, therefore, biochar has a broad application prospect in soil improvement[37, 38]. Application of water-soluble fertilizers improves soil quality and increases crop yields in alkaline soils[39].

37. Hossain MZ, Bahar MM, Sarkar B, Donne SW, Ok YS, Bolan N. Biochar and its importance on nutrient dynamics in soil and plant. Biochar. 2020; 2: 379-420.

https://doi.org/10.1007/s42773-020-00065-z.

38. Yuan P, Wang J, Pan Y, Shen B. Review of biochar for the management of contaminated soil: Preparation, application and prospect. Science of the total environment. 2019; 659: 473-490. 

https://doi.org/10.1016/j.scitotenv.2018.12.400.

39. He K, Xu Y, He G, Zhao X, Wang C, Li S, Zhou G, Hu R. Combined application of acidic biochar and fertilizer synergistically enhances Miscanthus productivity in coastal saline-alkaline soil. Science of The Total Environment. 2023; 164811. 

https://doi.org/10.1016/j.scitotenv.2023.164811.

Best regards

Haibo Chang

---

## [Decision Letter · Decision Letter 2]

14 Jan 2024

PONE-D-23-28238R2The effect of modifier and a water-soluble fertilizer on two forages grown in saline-alkaline soilPLOS ONE

Dear Dr. Chang,

Thank you for submitting your manuscript to PLOS ONE. After careful consideration, we feel that it has merit but does not fully meet PLOS ONE’s publication criteria as it currently stands. Therefore, we invite you to submit a revised version of the manuscript that addresses the points raised during the review process.

We look forward to receiving your revised manuscript.

Kind regards,

Mehdi Rahimi, Ph.D.

Academic Editor

PLOS ONE

Journal Requirements:

Reviewers' comments:

Reviewer's Responses to Questions

**Comments to the Author**

1. If the authors have adequately addressed your comments raised in a previous round of review and you feel that this manuscript is now acceptable for publication, you may indicate that here to bypass the “Comments to the Author” section, enter your conflict of interest statement in the “Confidential to Editor” section, and submit your "Accept" recommendation.

Reviewer #2: All comments have been addressed

2. Is the manuscript technically sound, and do the data support the conclusions?

Reviewer #2: Yes

3. Has the statistical analysis been performed appropriately and rigorously? 

Reviewer #2: Yes

4. Have the authors made all data underlying the findings in their manuscript fully available?

Reviewer #2: Yes

5. Is the manuscript presented in an intelligible fashion and written in standard English?

Reviewer #2: Yes

6. Review Comments to the Author

Reviewer #2: Please replace soda saline-alkali soil with saline -alkali soil or sodic soil through out the manuscript. Because soda saline-alkali soils are same as sodic or saline -alkali soil. If author wish to explain it differently, please submit your response. The data in tabular form is minimal, as the table data give better idea of each treatments and their interaction with other factor. Hence can be included for better understanding.

7. PLOS authors have the option to publish the peer review history of their article (what does this mean?). If published, this will include your full peer review and any attached files.

Reviewer #2: **Yes: **Sanjay Singh Rathore

---

## [Author Response · Author response to Decision Letter 2]

19 Jan 2024

Manuscript ID: PONE-D-23-28238

TITLE: The effect of modifier and a water-soluble fertilizer on two forages grown in saline-alkaline soil

Thank you very much for returning to us the Reviewers’ comments and queries. We also thank the Reviewers for their time and effort in evaluating our work. Please find our response to the points raised by the Reviewers in blue; changes in the manuscript are highlighted in red. We hope the revised manuscript can be accepted for publication on PLoS One.

Reviewer 2 comments:

1. Please replace soda saline-alkali soil with saline -alkali soil or sodic soil through out the manuscript. Because soda saline-alkali soils are same as sodic or saline -alkali soil. If author wish to explain it differently, please submit your response. The data in tabular form is minimal, as the table data give better idea of each treatments and their interaction with other factor. Hence can be included for better understanding.

Thanks for your suggestion. I have replaced soda saline-alkali soil with saline -alkali soil through out the manuscript. The basic properties of saline-alkali soil and The nutrient content of the rice straw and the waste fungus chaff The relevant data have been tabulated in the supporting information.

Introduction revised in the manuscript:

 Keywords: saline-alkali soil, modifier, water-soluble fertilizers, alfalfa, leymus chinensis

The Songnen Plain with an area of 37,000 km2 is the largest saline-alkali land in China. Among the three major regions worldwide where salt and alkali are distributed, the Songnen Plain demonstrates the highest ecological degradation rate. 

The saline-alkali soil sample was collected in Erlong Town, Taonan City, Jilin Province, which belonged to Songnen Plain.

Best regards

Haibo Chang

---

## [Decision Letter · Decision Letter 3]

6 Feb 2024

The effect of modifier and a water-soluble fertilizer on two forages grown in saline-alkaline soil

PONE-D-23-28238R3

Dear Dr. Chang,

We’re pleased to inform you that your manuscript has been judged scientifically suitable for publication and will be formally accepted for publication once it meets all outstanding technical requirements.

Kind regards,

Mehdi Rahimi, Ph.D.

Academic Editor

PLOS ONE

Additional Editor Comments (optional):

Reviewers' comments:

Reviewer's Responses to Questions

**Comments to the Author**

1. If the authors have adequately addressed your comments raised in a previous round of review and you feel that this manuscript is now acceptable for publication, you may indicate that here to bypass the “Comments to the Author” section, enter your conflict of interest statement in the “Confidential to Editor” section, and submit your "Accept" recommendation.

Reviewer #2: All comments have been addressed

2. Is the manuscript technically sound, and do the data support the conclusions?

Reviewer #2: Yes

3. Has the statistical analysis been performed appropriately and rigorously? 

Reviewer #2: Yes

4. Have the authors made all data underlying the findings in their manuscript fully available?

Reviewer #2: Yes

5. Is the manuscript presented in an intelligible fashion and written in standard English?

Reviewer #2: Yes

6. Review Comments to the Author

Reviewer #2: The manuscript is now improved significantly. As the authors have addressed all the concern raised and incorporated these in revised manuscript. Hence may be accepted for publication.

7. PLOS authors have the option to publish the peer review history of their article (what does this mean?). If published, this will include your full peer review and any attached files.

Reviewer #2: **Yes: **Sanjay Singh Rathore

---

## [Editor Report · Acceptance letter]

19 Feb 2024

PONE-D-23-28238R3 

PLOS ONE

Dear Dr. Chang, 

I'm pleased to inform you that your manuscript has been deemed suitable for publication in PLOS ONE. Congratulations! Your manuscript is now being handed over to our production team.

Kind regards, 

on behalf of

Associate Prof. Mehdi Rahimi 

Academic Editor

PLOS ONE